# Asymptomatic Strongyloidiasis among Latin American Migrants in Spain: A Community-Based Approach

**DOI:** 10.3390/pathogens9060511

**Published:** 2020-06-24

**Authors:** Violeta Ramos-Sesma, Miriam Navarro, Jara Llenas-García, Concepción Gil-Anguita, Diego Torrús-Tendero, Philip Wikman-Jorgensen, Concepción Amador-Prous, María-Paz Ventero-Martín, Ana-María Garijo-Sainz, María García-López, Ana-Isabel Pujades-Tárraga, Cristina Bernal-Alcaraz, Antonio Santonja, Pedro Guevara-Hernández, María Flores-Chávez, José-María Saugar, José-Manuel Ramos-Rincón

**Affiliations:** 1Internal Medicine Service, HLA Inmaculada Hospital, 18004 Granada, Spain; violeta.ramos@grupohla.com; 2Public Health, Science History and Gynecology Department, Universidad Miguel Hernández, 03550 Sant Joan d’Alacant, Alicante, Spain; navarro_mirbel@gva.es; 3Internal Medicine Service, Vega Baja Hospital-FISABIO, Orihuela, 03314 Alicante, Spain; jllenas@umh.es (J.L.-G.); garcia_marialop@gva.es (M.G.-L.); cristinabernalalcaraz@hotmail.com (C.B.-A.); pegueher@gmail.com (P.G.-H.); 4Clinica Medicine Department, University Miguel Hernández de Elche, 03550 Sant Joan d’Alacant, Alicante, Spain; 5Internal Medicine Service, Marina Baixa Hospital, La Vila Joiosa, 03570 Alicante, Spain; gil_conang@gva.es (C.G.-A.); amador_con@gva.es (C.A.-P.); garijo_ana@gva.es (A.-M.G.-S.); anabelpujades@gmail.com (A.-I.P.-T.); santonja_ant@gva.es (A.S.); 6Internal Medicine Service, General University Hospital of Alicante-ISABIAL, 03550 Alicante, Spain; torrus_die@gva.es; 7Department of Parasitology, Universidad Miguel Hernández de Elche, 03550 Alicante, Spain; 8Internal Medicine Service, University Clinical Hospital Sant Joan d’Alacant-FISABIO, 03550 Alicante, Spain; wikman_phi@gva.es; 9Microbiology Service, General University Hospital of Alicante -ISABIAL, 03010 Alicante, Spain; ventero_marmar@gva.es; 10Foundation Mundo Sano, 28046 Madrid, Spain; maria.flores@mundo.sano.org; 11Parasitology Service, National Center of Microbiology, 28222 Madrid, Spain; jmsaugar@isciii.es

**Keywords:** Strongyloidiasis, *Strongyloides**stercoralis*, Chagas disease, Central and South America, Community-based intervention, migrants

## Abstract

*Strongyloides stercoralis* infection is frequently underdiagnosed since many infections remain asymptomatic. Aim: To estimate the prevalence and characteristics of asymptomatic *S. stercoralis* infection in Latin American migrants attending a community-based screening program for Chagas disease in Spain. Methodology: Three community-based Chagas disease screening campaigns were performed in Alicante (Spain) in 2016, 2017, and 2018. Serological testing for *S. stercoralis* infection was performed using a non-automatized IVD-ELISA detecting IgG (DRG Instruments GmbH, Marburg, Germany). Results: Of the 616 migrants from Central and South America who were screened, 601 were included in the study: 100 children and adolescents (<18 years of age) and 501 adults. Among the younger group, 6 participants tested positive (prevalence 6%, 95% confidence interval [CI] 2.5% to 13.1%), while 60 adults did so (prevalence 12%, 95% CI 9.3% to 15.3%). *S. stercoralis* infection was more common in men than in women (odds ratio adjusted [ORa] 2.28, 95% CI 1.289 to 4.03) and in those from Bolivia (ORa 2.03, 95% CI 1.15 to 3.59). Prevalence increased with age (ORa 1.02, 95% CI 0.99 to 1.05). In contrast, a university education had a protective effect (ORa 0.29, 95% CI 0.31 to 0.88). Forty-one (41/66; 62.1%) of the total cases of *S. stercoralis* infection were treated at the health care center. Positive stool samples were observed in 19.5% of the followed-up positive cases. Conclusion: Incorporating serological screening for *S. stercoralis* into community-based screening for Chagas disease is a useful intervention to detect asymptomatic *S. stercoralis* infection in Central and South American migrants and an opportunity to tackle neglected tropical diseases in a transversal way. The remaining challenge is to achieve patients’ adherence to the medical follow-up.

## 1. Introduction

Strongyloidiasis, the human disease caused by infection with *Strongyloides stercoralis* [1], is a soil-transmitted helminthiasis with at least 30 to 100 million people infected globally [2]. The real prevalence is probably underestimated because of the low sensitivity of traditional diagnostic methods [3]. The medical importance of this infection resides in its capacity to remain clinically asymptomatic for years, persisting in the host for a lifetime as a result of autoinfection. Although it often causes nothing more than mild gastrointestinal, respiratory or cutaneous symptoms, alterations in the infected person’s immune system can lead to hyperinfection syndrome, with dissemination of large numbers of larvae from gastrointestinal tracts and lungs to ectopic sites due to accelerated larval reproduction. This can result in severe systemic bacterial infections that may lead to multiorgan failure and death [4,5,6]. Hyperinfection and disseminated strongyloidiasis can be fatal, especially among immunocompromised individuals, with a reported mortality up to 62% [7,8,9].

The migrant population in Spain has increased significantly during the last fifteen years. In 2018, there were 6 million people from *S. stercoralis*-endemic countries living in Spain. This group includes 1.8 million people from South America and 500,000 from Central America and the Caribbean [10]. Latin America has been described as a region with high prevalence of chronic strongyloidiasis [11,12]. Many cases remain undetected for years in migrants’ destination country because of their asymptomatic or mildly symptomatic nature [6,13,14].

Diagnosing asymptomatic strongyloidiasis would enable its treatment, potentially preventing subsequent development of hyperinfection or disseminated strongyloidiasis [6,14]. Different authors in the United States and Europe have described experiences diagnosing asymptomatic Chagas disease in the community [15,16,17,18]. To our knowledge, however, there are no published strategies describing the diagnosis of asymptomatic strongyloidiasis through community-based interventions in high income-countries, apart from those in Madrid and Alicante (Spain) [19].

The burden and complexity of this silent and potentially lethal infection make it necessary to actively search for people with strongyloidiasis in order to offer them specific treatment and medical follow-up when indicated. The aim of this study is to describe the results of opportunistic serological screening for strongyloidiasis in Latin American migrants attending a community-based screening program for Chagas disease in Spain.

## 2. Material and Methods

### 2.1. Study Population

This was a cross-sectional study conducted during three community screening campaigns for Chagas disease and strongyloidiasis, involving three centers in the province of Alicante (Spain) and taking place on 31 January 2016, 30 May 2017, and 28 October 2018. Alicante is a Spanish province located in south-eastern Spain. It has a population of 1.8 million people, of whom about 13.8% are migrants. Around 48,700 of these come from Central and South America [20]. Callosa d’en Sarrià is a city located about 40 km north of Alicante city, and Orihuela is 60 km to the south (Figure 1).

The 2016 and 2017 campaigns were organized by a multidisciplinary team belonging to ISABIAL/FISABIO Research Foundation, the General University Hospital of Alicante, and the Foundation Mundo Sano, and they were performed in the abovementioned hospital. The 2018 campaign was organized by a multidisciplinary team belonging to the ISABIAL/FISABIO Research Foundation and three hospitals in Alicante province, (Alicante city, Villajoyosa and Orihuela) and was performed simultaneously in the General University Hospital of Alicante and two primary healthcare centers (Callosa d’en Sarrià and Álvarez de la Riva in Orihuela).

The inclusion criteria were: all adult participants who attended the community screening campaigns for Chagas disease and strongyloidiasis signed the informed consent, and for non-adult participants their parents or tutors gave their written consent. We excluded adults born in Spain, participants who did not undergo serological testing, and participants in whom the serum specimen was insufficient.

Trained Latin American community health workers promoted the event for a few weeks before the date of its celebration. Latin American migrants were informed through the dissemination of informative leaflets and posters at key locations (NGOs, small businesses, restaurants, shops, markets, etc.); informative talks at local Latin American meeting points and community events (e.g., sporting events); and social networks and media aimed at the Latin American population (radio and newspapers).

### 2.2. Questionnaires

Before the blood extraction, all participants completed a questionnaire eliciting epidemiological data. An additional questionnaire, used primarily to assess the level of knowledge about Chagas disease, also contained a question about strongyloidiasis: *“Have you ever heard of the Strongyloides parasite or strongyloidiasis?”* Medical doctors and medical or nursing students were always available for assistance.

### 2.3. Procedure: Serological Methods

Detection of *Strongyloides* IgG antibodies was performed using the *Strongyloides* IgG IVD-ELISA kit (DRG Instruments GmbH, Marburg, Germany). It includes microtiter wells coated with a soluble fraction of *S. stercoralis* L3 filariform larval antigen. The test was considered positive if the index (optical density [OD] measure of the sample divided by the cutoff value) was more than 1.1. The result was considered borderline if the index was between 1.0 and 1.1. Analyses were performed in the Parasitology Department of the National Center for Microbiology-Health Institute Carlos III (DP-NCM-ISCIII) in Madrid.

### 2.4. Follow-Up of Participants with Positive Serology 

Participants with positive serology for strongyloidiasis were traced and offered specialized outpatient clinical management by a medical doctor at their refence hospital. 

Their full medical history was taken, and other tests were done to complete the study according to their physician’s criteria. Tests included one or several samples to examine *S. stercoralis* in feces (Baermann technique, agar plate culture, or a molecular diagnostic method (real-time polymerase chain reaction [RT-PCR]), depending of the protocol in each hospital, and a complete blood test with eosinophil count and IgE levels [21,22,23]. The *Strongyloides* RT-PCR was performed in the DP-NCM-ISCIII, following the same methodology described by Saugar et al. 2015 [23].

The cases were treated with one or two doses of ivermectin (200 μg/kg/day) [24]. Because treatment protocols are not standardized in the study centers, some patients were treated on day 1 and 14, while others took the drug on two consecutive days.

Cure was defined as: (1) negative stool examination and decrease of initial eosinophil count by at least half, and/or (2) seroconversion or at least half of initial OD/relative light units values for ELISA tests, six months after treatment completion [25].

### 2.5. Statistical Analysis

Categorical data are presented as absolute and relative frequencies, and continuous variables as medians and interquartile ranges (IQRs). Lower and upper limits of the 95% confidence interval (CI) for prevalence were calculated following the methods described by Newcombe et al. The chi-square test or Fisher’s exact tests were used, as appropriate, to compare the distribution of categorical variables, and the Mann-Whitney U test was used for continuous variables. Associations were measured using the odds ratio (OR) with a 95% CI. Following univariable analyses, we fit a multivariable multiple regression model using a forward stepwise approach to identify variables independently associated with a specific infection. Results were considered statistically significant if the two-tailed *p* value was less than 0.05. SPSS Statistics for Windows, Version 21.0 (IBM Corp., Armonk, NY, USA) was used for statistical analysis.

### 2.6. Ethical Considerations

The STROBE statement guidelines were followed in the conducting and reporting of the study (Appendix A). Procedures were performed in accordance with the ethical standards set out in the Declaration of Helsinki, as revised in 2013. All three community-based screening campaigns were approved by the General University Hospital of Alicante Ethics Committee (Valencian Healthcare Agency; ref: CEIC PI2015 /16 and ref. CEI PI2018/035). All participants signed written informed consent.

## 3. Results

Of the 616 people who took part in the screening, 601 were included in the study. Figure 2 shows the participant flow chart. 

Of the 601 participants, 128 were screened in 2016, 114 in 2017, and 359 in 2018. There were 100 (16.6%) children and adolescents (<18 years) and 501 (83.9%) adults (≥18 years).

### 3.1. S. stercoralis Infection in Children and Adolescents

The population screened aged under 18 years included 57 girls (57%), and the sample had a median age of 11 years; 74% were born in Spain (Table 1). Six tested positive for *S. stercoralis* infection (prevalence 6%, 95% CI 2.5% to 13.1%). The median values of ELISA titers was 1.3 (IQR 1.7 to 3.5; range 1.2 to 7.8), and these participants were younger than those who were not infected (median 5 vs 12 years old, OR 0.58, 95% CI 0.39 to 0.85). All six infected children were born in Spain, but their mothers were from Ecuador (n = 3), Bolivia (n = 2), and Colombia (n = 1). All had traveled to their parents’ country of origin in the last three years. 

### 3.2. S. stercoralis Infection in Adult Participants 

Of the 501 adult participants, most were women (n = 303, 60.5%), and the median age was 41 years. The main countries of birth were Bolivia (40.3%) and Ecuador (37.5%), and they had been in Spain for a median time of 11 years. Only 7.9% had heard of *Strongyloides* parasites (Table 2). In total, 60 were positive for *S. stercoralis* infection (prevalence 12%, 95% CI 9.3% to 15.3%). The median value ELISA titers was 4.8, (IQR 1.9 to 10.5; range 1.1 to 22.3).

Table 2 shows the epidemiological characteristics of participants with and without *S. stercoralis* infection. Strongyloidiasis was more common in men and in those born in Bolivia but less common in those who had completed university studies. Prevalence increased slightly with age. In the multivariable analysis *S. stercoralis* infection was associated with male sex (adjusted OR 2.28, 95% CI 1.9 to 4.03) and birthplace in Bolivia (OR 2.03, 95% CI 1.15 to 3.58). Completion of university studies was a protective factor (adjusted OR 0.29, 95% CI 0.10 to 0.88). 

Of the 60 participants with *S. stercoralis* infection, 14 (23.3%; 95% CI 13.8% to 36.2%) had positive *Trypanosoma cruzi* serology. All were from Bolivia. The prevalence of strongyloidiasis-Chagas co-infections in Bolivians was 11.4% (95% CI 7.5% to 16.8%).

### 3.3. Follow-Up of Participants with S. stercoralis Positive Serology

Twenty-five of the 66 (37.5%) total positive participants were lost to follow-up (Table 3). Of the 41 infected patients who did present to the outpatient clinic at least once, a fecal parasitological analysis was done in 35; 8 (19.5%) samples were positive for *S. stercoralis* larvae. Eosinophilia was detected in 16/27 (59.2%) patients, and IgE values were indicative of infection in 9/10 (90%). Treatment was offered to 28 patients; the rest did not come back to the following appointments or were checked once as outpatients. Twenty-eight patients were treated, of whom 12 were confirmed as cured. Four were lost to follow-up and 12 were still under follow-up at the time of writing. 

## 4. Discussion

Global migration from *S. stercoralis*-endemic regions has increased the potential individual and public health impact of this parasitic NTD [26]. Active searches for susceptible people have proven effective in increasing its diagnosis [13,27,28], enabling prompt treatment and prevention of further transmission and hyperinfection syndrome [26,27]. 

*S. stercoralis* infection is estimated to have a prevalence of between 7% and 40% among migrants, depending on the survey and patients’ country of origin [14,29]. In a recent meta-analysis of the prevalence of strongyloidiasis in migrants from endemic areas who reside in Spain, the authors found an overall prevalence of 14% in Latin Americans [30]. Our study showed a lower prevalence than that (6% in children and adolescents and 12% in adults; 11% overall), but a similar proportion as another study performed in Spain, where the prevalence was 10.9% in patients from South America [3]. 

We found significant differences in strongyloidiasis prevalence by sex, an observation that does not match the results of other articles [29]. On the other hand, we also found a significant difference in relation with age, which is consistent with other studies performed in endemic areas [9,31].

The prevalence of co-infection between *T. cruzi* and *S. stercoralis* is not negligible [29,32]. In our study, 14 of 60 adults (23.3%, or around one in five cases) with *S. stercoralis* infection had also positive *T. cruzi* serology. Our results are similar to those reported by a Spanish Collaborative Network, where the co-infection prevalence was 21% [26]. Moreover, Salvador et al. [33] have found that patients with Chagas disease and positive *S. stercoralis* serology had a higher proportion of DNA of *T. cruzi* by RT-PCR in peripheral blood than those with negative *Strongyloides* serology, which reflects the potential immunomodulatory effects of *S. stercoralis* in *T. cruzi* co-infected patients. Both of the infections are neglected diseases, so co-infection is normally underestimated in published studies [32,33,34].

Regarding strongyloidiasis in children, *S. stercoralis* was detected in 6% of the participants aged under 18 years. In endemic countries with poor sanitation or inadequate water supply, *S. stercoralis* infection is common in this age group [1,2] and causes nutritional impairment, with the important consequences that this implies in children [35]. However, few studies have specifically analyzed the relevance of serology for the diagnosis of chronic *S. stercoralis* infections in children in non-endemic countries [36]. In our study, most of the children screened were born in Spain, but at least one of their parents came from Latin America. All six children with positive *S. stercoralis* serology had traveled to their parents’ countries in the recent past, so they should be considered migrants visiting friends and relatives. This community-based approach poses an opportunity to perform health education about the global health risks and preventative measures before, during, and after travelling [37], and it may lead to a reduction in the number of future disseminated strongyloidiasis cases. The low titers shown on the ELISA may be due to cross-reactivity.

Regarding knowledge of strongyloidiasis in the community, only 8% of the adult respondents had heard of the parasite. This result is not surprising, as human strongyloidiasis has been repeatedly pointed out as a largely neglected entity [4], and one characteristic of NTDs is the low awareness of them. This lack of awareness is not limited to the populations at risk but also extends to future healthcare professionals [38].

The determination of IgG by ELISA is a good technique for the screening of chronic *S. stercoralis-*infected patients [29,36]. The confirmatory diagnosis of *S. stercoralis* infection is based on the detection of larvae in stool samples, but in most chronic asymptomatic patients, the intestinal worm load is very low, and the output of larvae is minimal and irregular [36,39]. In our study, 19% of the screen-positive samples showed confirmation of larvae in stool culture (stool direct observation) or by detection of parasite DNA. This result is concordant with results obtained in other studies performed among migrants living in Europe, where less than 20% of the patients with positive serology have positive results by stool microscopy [36,39]. There are differences in the sensitivity of these methods i.e., examining *S. stercoralis* in feces by parasitological methods (Baermann technique or agar plate culture) and a molecular diagnostic method. The sensitivity of RT-PCR is higher than parasitological methods only, including serology [40].

We had a high rate of attrition in screened positive patients. Only two-thirds attended the first clinical visit, and less than half returned to receive strongyloidiasis treatment. The rate of loss to follow-up increases with the interval between the screening and the medical appointment. Lack of presentation to the consultation also increases with subsequent follow-up visits. This conclusion is similar to that of other studies on Chagas disease, where only about half of patients diagnosed after an additional test presented for treatment [41,42]. The reasons are diverse: difficulties with contacting patients, structural barriers, high mobility of the migrant population in and outside of the country, and the asymptomatic nature of *S. stercoralis* infection, among others [43]. Strategies to optimize retention must be an essential component of future interventions and could include administering treatment at the first visit, making sure sufficient contact information is provided, using telemedicine to minimize visits to the hospital, tracing patients with high mobility, and improving communication between primary and hospital healthcare and between hospitals of different geographical settings. Nevertheless, some barriers will be quite difficult to overcome, such as the structural ones, especially regarding the host country [44]

About half of our participants treated for *S. stercoralis* infection were lost to follow-up or had not yet completed the six-month follow-up period at the time of writing. This result is comparable to other studies [39]. Most cases that had completed follow-up by the end of the study had been cured of chronic *S. stercoralis* infection. In that regard, *S. stercoralis* serology is useful for post-treatment follow-up, as also observed by several research studies in immigrant populations in Europe and North America [39,44].

Given the low adherence to treatment and follow-up of our participants, offering *S. stercoralis* treatment and follow-up at a primary healthcare level is warranted. A recent review about different strategies for *S. stercoralis* screening and treatment among migrants is pointing in that direction [43]. 

This study has several limitations. First, it was a cross-sectional study conducted in Alicante, so the findings might not be generalizable to populations that are very different from our sample. Second, most of the participants were from Bolivia, because at its inception, our community-based campaign was designed to diagnose Chagas disease; diagnosis of *S. stercoralis* was an added value for its participants. Thus, the main target population was Bolivian migrants, who make up more than 80% of the Chagas patients in Spain [45]. There is less representation from other countries of origin like Colombia, Argentina, or Paraguay, and none from the Caribbean. Third, there were some difficulties in contacting patients, making appointments, and ensuring their presentation to the clinic. Furthermore, the questionnaire could have been more specific, asking more details about recent travel, including the length and specific places that were visited, in order to collect precise epidemiological data and draw more robust conclusions among the participants who had visited their home country to visit relatives and friends. Finally, there was no uniform protocol for action in the different healthcare centers where treatment and medical follow-up took place.

## 5. Conclusions

In our study we found an 11% prevalence of strongyloidiasis among the migrants, which is similar to other studies performed in Spain, despite our limitations and the regional nature of our campaign. Community-based interventions such as the screening campaigns described in this manuscript, which actively search for patients with chronic *S. stercoralis* infection, are scarce. The strategy of linking *S. stercoralis* screening with community-based screening for Chagas disease is a useful, convenient, and efficient intervention to detect asymptomatic strongyloidiasis in Latin American migrants. Moreover, it represents an opportunity to tackle NTDs in a transversal way.

The European Centre for Disease Prevention and Control recommends screening newly arrived migrants for strongyloidiasis [46,47], particularly immunosuppressed individuals [48], given the potential individual morbidity and mortality. However, this is not widely performed. It is important to also address the barriers to proper follow-up of positive cases in future campaigns, as only two-thirds of infected participants finally received care for the disease. Larger and more in-depth studies are needed to confirm the benefits of community-based screening for chronic *S. stercoralis* infection and the economic impact of preventing a severe symptomatic infection. 

## Figures and Tables

**Figure 1 pathogens-09-00511-f001:**
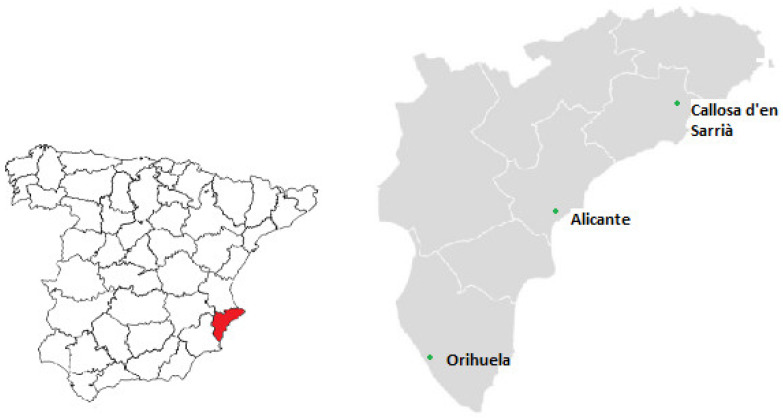
Map of Spain, with Alicante city, Callosa d’en Sarrià and Orihuela.

**Figure 2 pathogens-09-00511-f002:**
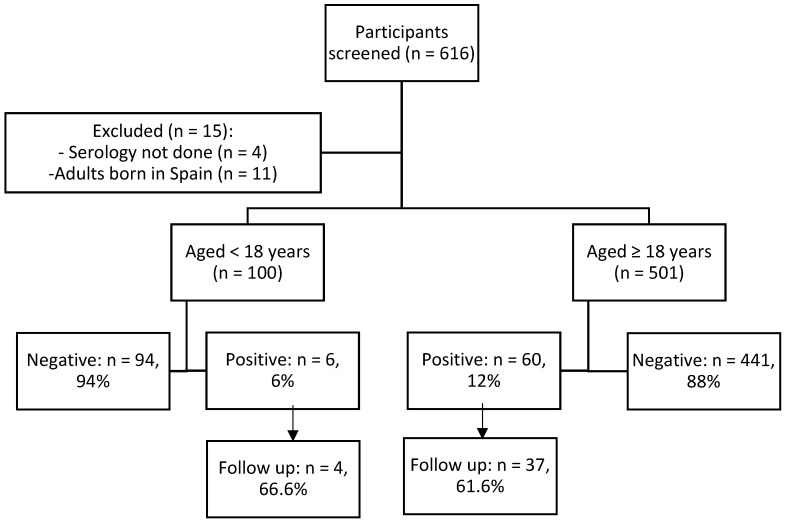
Participants flow chart.

**Table 1 pathogens-09-00511-t001:** Characteristics of children and adolescent participants with and without *S. stercoralis* infection.

	Total (n = 100)	*S. stercoralis* Infection (n = 6)	No Infection (n = 94)	OR (95% CI)	*p* Value
**Demographics**					
Boys, n (%)	43 (43)	4 (66.7)	2 (2.1)	2.82 (0.49–16.17)	0.23
Median age, years (IQR) (n = 69)	11 (9–14)	5 (4–6)	12 (10–14)	0.58 (0.39–0.85)	0.001
**Country of birth,** n (%)					
Spain *	74 (74)	6 (100)	68 (72.3)	NA	0.13
Bolivia	13 (13)	0	13 (13.8)	NA	0.99
Ecuador	9 (9)	0	9 (9.6)	NA	0.99
Argentina	2 (0)	0	2 (0.1)	NA	0.99
Uruguay	2 (2)	0 (0.0)	2 (0.1)	NA	0.99

IQR: interquartile range, OR: odds ratio; CI: confidence intervals, * Children born in Spain, whose mother comes from Latin America. Nationality of the mother: (positive/negative): Bolivia (n = 2/35), Ecuador (n = 3/28), Colombia (n = 1/6), Argentina (0/1), not available (n = 0/4).

**Table 2 pathogens-09-00511-t002:** Characteristics of adult participants with and without *S. stercoralis* infection.

	Total (n = 501)	*S. stercoralis* Infection (n = 60)	No Infection (n = 441)	OR (95% CI)	*p* Value	ORa (95% CI)	*p* Value
**Demographics**							
Men, n (%)	198 (39.5)	34 (56.7)	164 (37.2)	2.20 (1.28–3.81)	0.004	2.28 (1.29–4.03)	0.004
Median age, years (IQR) (n = 493)	41 (34–49)	44 (37–51)	41 (34–49)	1.02 (1.00–1.04)	0.046	1.02 (0.99–1.05)	0.064
**Education,** n (%) **(n = 478)**							
Primary school	128 (26.9)	21 (35.6)	107 (25.5)	1.61 (0.90–2.86)	0.10	-	
Secondary school	226 (55.6)	34 (57.6)	232 (55.4)	1.09 (0.63–1.90)	0.71	-	
University studies	84 (17.6)	4 (6.8)	80 (19.1)	0.30 (0.10–0.87)	0.02	0.29 (0.31–0.88)	0.029
**Country of birth,** n (%)							
Bolivia	202 (40.3)	33 (55)	169 (38.3)	1.96 (1.14–3.38)	0.013	2.03 (1.15–3.59)	0.014
Ecuador	188 (37.5)	19 (31.7)	169 (38.3)	0.74 (0.41–1.32)	0.32	-	
Colombia	65 (13.0)	6 (10.0)	59 (13.4)	0.71 (0.29–1.74)	0.46	-	
Argentina	13 (2.6)	1 (1.7)	12 (2.7)	0.60 (0.07–4.75)	0.63	-	
Brazil	7 (1.4)	0 (0.0)	7 (1.6)	NA	0.99	-	
Paraguay	6 (1.2)	0 (0.0)	6 (1.4)	NA	0.99	-	
Dominican Republic	4 (0.8)	0 (0.0)	4 (0.9)	NA	0.99	-	
Peru	4 (0.8)	0 (0.0)	4 (0.9)	NA	0.99	-	
Venezuela	4 (0.8)	0 (0.0)	4 (0.9)	NA	0.99	-	
Other *	8 (1.6)	1 (1.7)	7 (1.6)	1.01 (0.12–8.69)	0.99	-	
**Knowledge about disease,** n affirmative response (%)							
Have you ever heard of *Strongyloides* parasite or strongyloidiasis?	39 (7.9)	1 (1.7)	38 (8.7)	0.18 (0.02–1.35)	0.07	-	

IQR: interquartile range, OR: odds ratio; ORa: adjusted odds ratio; CI: confidence intervals, * Other countries: Nicaragua (n = 2), Uruguay (n = 2), Honduras (n = 2), Mexico (n = 2).

**Table 3 pathogens-09-00511-t003:** Follow-up after screening campaign in children and adults with *S. stercoralis* positive serology.

Variable	Total n/N (%)	Children n/N (%)	Adults n/N (%)
*S. stercoralis* infection	66/601 (11)	6/100 (6)	60/501 (11)
**Follow-up of patients**			
Available	41/66 (62.1)	4/6 (66.6)	37/60 (61.6)
Unavailable	25/66 (37.9)	2/6 (33.3)	23/60 (38.3)
**Reason for not following up**			
No phone contact	17/25 (68)	1/2 (50)	16/23 (69.5)
Current phone not available	7/25 (28)	1/2 (50)	6/23 (26.1)
Picked up the phone, but did not go to the appointment	1/25 (4)	0	1/23 (4.3)
**Follow-up for confirmation**			
Stool examination			
Negative	27/41(65.9)	2/4 (50)	23/37 (62.2)
Positive *	8/41 (19.5)	0	8/37 (21.6)
Not recovered	6/41 (14.6)		
Eosinophilia (>5% leukocyte or >500 eosinophils)	16/27 (59.2)	1/1 (100)	15/26 (57.7)
IgE > 100	9/10 (90)	1/1 (100)	8/9 (88.8)
**Outcome**			
Treatment	28/41 (68.3)	2/4 (50)	26/37 (70.2)
**Results of treatment ^†^**			
Cure	12/28 (42.9)	0	12/26 (46.2)
Ongoing	12/28 (42.9)	0	10/26 (38.4)
Lost to follow-up after treatment	4/28 (14.2)	2/4 (50)	4/26 (15.4)

* Real-time polymerase chain reaction (RT-PCR) in stool (n = 13), stool culture (n = 27), both RT-PCR and culture (n = 13). ^†^ All patients that finished treatment (12/12) were cured.

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
