# Peer review of "Asymptomatic Strongyloidiasis among Latin American Migrants in Spain: A Community-Based Approach"

_pathogens, 2020, doi:10.3390/pathogens9060511_

Round 1
Reviewer 1 Report
The paper of Violeta Ramos-Sesma et al “Asymptomatic strongyloidiasis among Latin American migrants in Spain: a community-based approach―Asymptomatic strongyloidiasis in Latin American migrants” presents interesting data concerning prevalence and characteristics of S. stercoralis infection in Latin American migrants in Alicante province (Spain). This is a classical epidemiological registration connected with statistical analysis of influencing factors. I think that it could be interesting for parasitologists/epidemiologists especially in Europe.
My comment concerns Material and methods. Namely, I have some doubts on description of methods for faeces examination. Authors wrote in part “Follow-up of participation with positive serology” that additional tests were perform: Baerman technique, agar plate culture, real-time PCR. However, we do not know exactly if all samples were examined by 3 methods, or some of sampeles were examined by Baerman, others with PCR… etc. it must be define clearly – there are differences in sensitivity of these methods what can be shortly discussed also in Discussion section.
Author Response
Reviewer 1
The paper of Violeta Ramos-Sesma et al “Asymptomatic strongyloidiasis among Latin American migrants in Spain: a community-based approach―Asymptomatic strongyloidiasis in Latin American migrants” presents interesting data concerning prevalence and characteristics of S. stercoralis infection in Latin American migrants in Alicante province (Spain). This is a classical epidemiological registration connected with statistical analysis of influencing factors. I think that it could be interesting for parasitologists/epidemiologists especially in Europe.
My comment concerns Material and methods. Namely, I have some doubts on description of methods for faeces examination. Authors wrote in part “Follow-up of participation with positive serology” that additional tests were perform: Baerman technique, agar plate culture, real-time PCR. However, we do not know exactly if all samples were examined by 3 methods, or some of samples were examined by Baerman, others with PCR… etc. it must be define clearly – there are differences in sensitivity of these methods what can be shortly discussed also in Discussion section.
Respond: Thanks for the comment.
In order to add clarity to the text, the new version now states (lines 133-136):
Tests could include one or several samples to examine S. stercoralis in feces (Baermann technique, agar plate culture, or a molecular diagnostic method (real-time polymerase chain reaction [RT-PCR]), depending of the protocol in each hospital”
Moreover, we have included in the discussion section in lines 310-312 as two sentences about differences in sensitivity of parasitological and molecular methods
There are differences in sensitivity of these methods examine S. stercoralis in feces by parasitological methods (Baermann technique or agar plate culture) and a molecular diagnostic method. The sensitivity of RT-PCR is higher than parasitological methods only and including serology[40].
Reviewer 2 Report
Epidemiological and descriptive work that is well structured, based on the analysis of the presence of S. stercoralis infection through immunological amplification of PCR.
About the amplification by PCR, the authors have not reflected the conditions of the reaction or the primers used, the reader is only directed to a reference without indicating that they have followed the same methodology or if there was any change
Although it is not indicated how disseminated strongyloidiasis is performed in this group of people. Or at least at theory
Author Response
Reviewer 2
Epidemiological and descriptive work that is well structured, based on the analysis of the presence of S. stercoralis infection through immunological amplification of PCR.
About the amplification by PCR, the authors have not reflected the conditions of the reaction or the primers used, the reader is only directed to a reference without indicating that they have followed the same methodology or if there was any change.
Respond: Thanks for the suggestions. According to the reviewer’s comment, we have included a sentence explaining the method used in the amplification by PCR. The new sentence states (line 136-138):
The Strongyloides RT-PCR was performed in the DP-NCM-ISCIII, following the same methodology described by Saugar et al. 2015 [23].
Although it is not indicated how disseminated strongyloidiasis is performed in this group of people. Or at least at theory.
Respond: Thanks for the comment. We have included a sentence regarding the reviewer’s suggestion. The original text was:
Although it often causes nothing more than mild gastrointestinal, respiratory or cutaneous symptoms, alterations in the infected person’s immune system can lead to hyperinfection syndrome, with dissemination due to accelerated larval reproduction. This can result in severe systemic bacterial infections that may lead to multiorgan failure and death [4][5][6]. Hyperinfection and disseminated strongyloidiasis can be fatal, especially among immunocompromised individuals, with a reported mortality up to 62% [7][8][9].
We have added (lines 65 and 66): “…of large numbers of larvae from gastrointestinal tracts and lungs to ectopic sites…”
Reviewer 3 Report
The study was designed to detect strongyloidiasis among migrants in the campaigns that aimed at detecting Chagas disease. The methods were appropriate and well-designed. Minor comment in table 1 as described below:
- In table 1, where in the no infection column, were 2 out of 94 uninfected patients boys? That would make 33.3% an inaccurate estimate.
What I found interesting is the infection in children born in Spain but have traveled to Latin America with their parents. Since the infection is asymptomatic and unlikely to cause significant harm unless the patient is immunocompromised, perhaps a recommendation to screen for the parasite among migrants who are immunocompromised would be an avenue to pursue in case patient adherence to ivermectin treatment does not work as an avenue.
Author Response
Reviewer 3
The study was designed to detect strongyloidiasis among migrants in the campaigns that aimed at detecting Chagas disease. The methods were appropriate and well-designed.
Minor comment
in table 1 as described below:
In table 1, where in the no infection column, were 2 out of 94 uninfected patients boys? That would make 33.3% an inaccurate estimate.
We thank the reviewer’s comment. The right percentage of uninfected boys is 2.1%. We have changed the table 1 accordingly.
What I found interesting is the infection in children born in Spain but have traveled to Latin America with their parents. Since the infection is asymptomatic and unlikely to cause significant harm unless the patient is immunocompromised, perhaps a recommendation to screen for the parasite among migrants who are immunocompromised would be an avenue to pursue in case patient adherence to ivermectin treatment does not work as an avenue.
We appreciate the reviewer’s comment. We have included information highlighting the importance of performing health education among VFRs, as helminthiasis among children causes nutritional impairments, and these community-based actions can reduce the future cases of strongyloidiasis among immunocompromised patients (following underlined text in lines 291 to 303 of the manuscript):
Regarding strongyloidiasis in children, S. stercoralis was detected in 6% of the participants aged under 18 years. In endemic countries with poor sanitation or inadequate water supply, S. stercoralis infection is common in this age group [1][2] and causes nutritional impairment, with the important consequences that this implies in children [35]. However, few studies have specifically analyzed the relevance of serology for the diagnosis of chronic S. stercoralis infections in children in non-endemic countries [36]. In our study, most of the children screened were born in Spain, but at least one of their parents came from Latin America. And all six children with positive S. stercoralis serology had traveled to their parents’ countries in the recent past, so they should be considered migrants visiting friends and relatives. This community-based approach poses an opportunity to perform health education about the global health risks and preventative measures before, during and after travelling [37], and it may lead to a reduction in the number of future disseminated strongyloidiasis cases. The low titers shown on the ELISA may be due to cross-reactivity